# ciBAR1 loss in mice causes laterality defects, pancreatic degeneration, and altered glucose tolerance

Eunice N Kim[1,2], Feng-Qian Li[2], Ken-Ichi Takemaru[1,2]

Bin/Amphiphysin/Rvs (BAR) domains are highly conserved domains found in all eukaryotes. BAR domain proteins form crescent-shaped dimers that sense and sculpt curved lipid membranes and play key roles in various cellular processes. However, their functions in mammalian development are poorly understood. We previously demonstrated that Chibby1-interacting BAR domain–containing 1 (ciBAR1, formerly known as FAM92A) localizes to the ciliary base and plays a critical role in ciliogenesis. Here, we report ciliopathy phenotypes of *ciBAR1*-KO mice. We found that ~28% of *ciBAR1*-KO mice show embryonic lethality because of randomized left–right asymmetry; the rest survive into adulthood with no gross morphological abnormalities. Histological assessments of ciliated tissues revealed exocrine pancreatic lesions. Although overall endocrine islet morphology appeared to be normal, *ciBAR1*-KO mice showed impaired glucose tolerance. Examination of ductal and islet cilia revealed that cilia number and length were significantly reduced in *ciBAR1*-KO pancreata. *ciBAR1*-KO MEFs also exhibited ciliary defects. Our findings indicate that ciBAR1 plays a critical role in ciliogenesis depending on the tissue and cell type in mice.

## Introduction

The Bin/Amphiphysin/Rvs (BAR) domain consists of three extended α-helices arranged antiparallel to each other and is evolutionarily conserved from yeast to human (Salzer et al, 2017; Carman & Dominguez, 2018; Nishimura et al, 2018; Simunovic et al, 2019). BAR domain–containing proteins form crescent-shaped dimers that bind to phosphoinositide-rich membranes to generate positive or negative curvatures. Based on their structural properties, BAR domains are broadly categorized into the BAR/N-BAR, F-BAR, and I-BAR subfamilies. N- and F-BAR domains induce a different range of positive membrane curvatures, whereas I-BAR domains generate negative membrane curvatures. There are over 70 family members of the BAR domain superfamily encoded in the human genome (Salzer et al, 2017; Carman & Dominguez, 2018). In vitro, BAR-domain proteins display the ability to reshape spherical liposomes into elongated tubules. Through influencing membrane morphology and dynamics, they are involved in a variety of important biological processes such as endocytosis, vesicle fusion and fission, cytokinesis, and cell migration. The in vivo functions of BAR-domain proteins in mammals, however, remain poorly understood.

Cilia are evolutionarily conserved microtubule-based organelles that project from the apical surface of many different cell types and play key roles in embryonic development and adult life in various organisms (Wallmeier et al, 2020; Derderian et al, 2023; Mill et al, 2023; Hilgendorf et al, 2024). Cilia are broadly divided into two types: nonmotile primary cilia and motile multicilia. The primary cilium is composed of a 9 + 0 microtubule structure and serves as a sensory organelle that responds to chemical and mechanical cues in the environment. Multicilia are comprised of a 9 + 2 microtubule structure and generate directional fluid flow on the cell surface through coordinated ciliary beating. During the G0/G1 phase of the cell cycle, cilia are assembled from the basal body, a derivative of the mother centriole. The older mother centriole in each centrosome harbors accessory structures such as subdistal and distal appendages that are absent on the younger daughter centriole (Kumar & Reiter, 2021; Zhao et al, 2023). The subdistal appendages play critical roles in microtubule nucleation and anchoring, whereas the distal appendages are important for membrane attachment in the process of vesicle recruitment and basal body docking to the plasma membrane. Cilium extension and maintenance rely on intraflagellar transport (IFT), a bidirectional transport of protein particles driven by kinesin and dynein motors along the microtubule axoneme (Lechtreck, 2022; Mill et al, 2023).

Dysfunction of cilia is associated with genetically heterogeneous multisystem disorders collectively known as ciliopathies (Wallmeier et al, 2020; Mill et al, 2023). Defective primary cilia lead to a wide spectrum of diseases including polycystic kidney disease, Bardet–Biedl syndrome (BBS), and Joubert syndrome (JBTS). Their clinical features include laterality defects, retinal degeneration, intellectual disability, skeletal abnormalities, and cystic diseases of the kidney, liver, and pancreas. Impaired function of motile multicilia causes primary ciliary dyskinesia. Clinical manifestations of primary ciliary

---

[1]Molecular and Cellular Biology Graduate Program, Stony Brook University, Stony Brook, NY, USA    [2]Department of Pharmacological Sciences, Stony Brook University, Stony Brook, NY, USA

Correspondence: ken-ichi.takemaru@stonybrook.edu

dyskinesia include recurrent respiratory infections, reduced fertility, laterality defects, and in rare instances, hydrocephalus.

In the pancreas, the exocrine compartment, which occupies >95% of the organ, consists of ductal cells and acinar cells that produce digestive enzymes, whereas the remaining tissue consists of endocrine islets of Langerhans containing five different cell types ($\alpha$, $\beta$, $\delta$, $\varepsilon$, and PP/F) that secrete hormones to regulate blood glucose levels. Primary cilia are only found on ductal and endocrine cells but not on acinar cells (Cano et al, 2004; Cano et al, 2006; diIorio et al, 2014; Lodh et al, 2014; Hughes et al, 2020; Cyge et al, 2021). Pancreatic lesions such as exocrine degeneration, acinar-to-ductal metaplasia, cysts, fibrosis, and pancreatitis have been reported in ciliopathy patients (van Asselt et al, 2013; Lodh et al, 2014) and mouse models (Lu et al, 1997; Cano et al, 2004, 2006; Augereau et al, 2016; Cyge et al, 2021). The underlying mechanisms of the pathologies at the molecular and cellular level remain to be elucidated. However, it is thought that dysfunction of primary cilia in the ducts results in impaired sensing of luminal flow of the digestive juice and subsequent obstruction of pancreatic ducts. This causes the leakage of digestive enzymes and the death of surrounding acinar cells.

Previously, we identified the Chibby1 (Cby1)-interacting BAR domain–containing protein 1 (ciBAR1, formerly known as FAM92A) as a novel binding partner for the basal body protein Cby1 using tandem affinity purification (Li et al, 2016). Cby1 is a conserved, small coiled-coil protein that localizes to the ciliary base and plays a critical role in ciliogenesis (Voronina et al, 2009; Love et al, 2010; Burke et al, 2014). Cby1-KO mice show ciliopathy phenotypes, such as upper airway infection, polycystic kidneys, and exocrine pancreatic degeneration and hydrocephalus and polydactyly at low frequencies (Voronina et al, 2009; Love et al, 2010; Lee et al, 2014; Cyge et al, 2021). In the pancreas of Cby1-KO mice, progressive acinar cell degeneration was observed after birth, resulting in severe pancreatic exocrine atrophy; the architecture of the endocrine islets, however, appeared relatively normal (Cyge et al, 2021). ciBAR1 (33 kD) and its paralog ciBAR2 (35 kD) each contains a single BAR domain where they share 52% amino acid identity and 75% similarity (Li et al, 2016). Although ciBAR1 is ubiquitously expressed in mice and humans, ciBAR2 expression is restricted to multiciliated cells, such as airway and ependymal ciliated cells, in both organisms. We previously reported that ciBAR proteins form homo- and heterodimers and exist as a complex with Cby1 (Li et al, 2016). The ciBAR/Cby1 complex is recruited to the distal region of basal bodies through Cby1's interaction with the distal appendage protein CEP164, where they play critical roles in ciliogenesis (Burke et al, 2014; Li et al, 2016). More recently, we demonstrated that ciBAR1-KO male mice show severe fertility defects because of sharply kinked sperm tails (Hoque et al, 2024). During spermiogenesis, ciBAR1 forms a complex with Cby3, a Cby1 paralog specifically expressed in the testis, and localizes to the curved membrane invagination proximal to the septin-based ring structure known as the annulus at the base of sperm flagella. Importantly, Cby3/ciBAR1 binding increases local membrane stiffness, which is essential for the precise positioning of the annulus along the sperm flagellum. However, the ciliary role of ciBAR1 in other tissues and cell types remains unknown.

In this study, we have characterized ciliopathy phenotypes of ciBAR1-KO mice and found that loss of ciBAR1 results in left–right laterality defects, exocrine pancreatic degeneration, and altered glucose metabolism because of defective primary cilia. In summary, our findings indicate that membrane-binding ciBAR1 is essential for ciliogenesis in mediating mammalian embryonic development and tissue homeostasis.

# Results

## Loss of ciBAR1 results in partial embryonic lethality associated with defects in the determination of left–right body asymmetry

Our recent work revealed that ciBAR1-KO mice are present at lower-than-expected Mendelian ratios (18%) at postnatal day 7 (P7) (Table 1) (Hoque et al, 2024), suggesting that loss of ciBAR1 may cause embryonic lethality. To investigate this possibility, we performed timed matings between heterozygotes and assessed the genotypes and phenotypes of the resulting litters at embryonic day (E) 9.5. A total of 65 embryos from 10 females were analyzed at E9.5 (Table 1). The E9.5 embryos examined did not show a deviation from the Mendelian distribution ($\chi^2$ = 1.4790; $P$ = 0.5172). By this stage, all WT embryos (n = 15) had undergone proper axial rotation with the tail pointing to the right side of the body (Fig 1). However, we noticed that 5 out of 20 ciBAR1-KO embryos (25%) exhibited reversal of embryo turning with the tail pointing to the left side, indicative of randomized left–right asymmetry (Fig 1) because of defective motile cilia in the embryonic node. In addition, 2 out of 30 (7%) of ciBAR1 heterozygous embryos showed reversed embryo turning, indicating occurrence of haploinsufficiency at low frequencies. We did not notice any abnormal situs in surviving ciBAR1-KO mice at P7 and adult. Taken together, these findings suggest that loss of ciBAR1 is associated with laterality defects likely because of impaired nodal cilia, leading to embryonic lethality.

## ciBAR1-KO mice exhibit pancreatic exocrine lesions but normal kidney histology

Defective primary cilia are associated with a wide range of genetically heterogeneous, multisystem diseases known as ciliopathies, including cystic lesions of the kidneys and the pancreas (Wallmeier et al, 2020; Mill et al, 2023). ciBAR1-KO mice in general weighed significantly less than their WT counterparts: 22.04 ± 0.6792 g (SEM) for WT versus 19.60 ± 0.6502 g (SEM) for ciBAR1 KO (Fig 2A). To investigate whether ciBAR1-KO mice show any ciliopathy phenotypes, we performed histological analyses of both kidneys and pancreata from adult mice. ciBAR1-KO mice did not show a significant difference in pancreas weights normalized to body weight compared with their WT counterparts: 8.988 ± 0.4856 mg/g (SEM) for WT versus 9.070 ± 0.8055 mg/g (SEM) for ciBAR1 KO (Fig 2B). However, inspection of periodic acid–Schiff (PAS)–stained pancreatic sections revealed that like Cby1-KO mice (Cyge et al, 2021), ciBAR1-KO mice consistently show common features of acinar cell degeneration such as disorganized acini, lipomatosis (Fig 2C, dotted line), abundant ducts (Fig 2C, arrowheads), fibrosis (Fig 2D, arrowheads), and infiltration of inflammatory cells (Fig 2E).

**Table 1.  Numbers of offspring from *ciBAR1* heterozygous intercrosses for E9.5 and postnatal day 7 (P7).**

| Age | | Genotype | | | Total | |
|-----|-----|-----|-----|-----|-----|-----|
| | | +/+ | +/− | −/− | | |
| E9.5 | Observed number | 15 | 30 | 20 | 65 | |
| | Expected number | 17 | 35 | 17 | | $\chi2 = 1.4790$ |
| | Expected ratio | 0.25 | 0.5 | 0.25 | | $P = 0.5172$ (ns) |
| | Actual ratio | 0.22 | 0.43 | 0.29 | | |
| P7 | Observed number | 73 | 160 | 51 | 284 | |
| | Expected number | 71 | 142 | 71 | | $\chi2 = 7.9718$ |
| | Expected ratio | 0.25 | 0.50 | 0.25 | | $P = 0.0186$ (*) |
| | Actual ratio | 0.26 | 0.56 | 0.18 | | |

The proportion of mice for each genotype was compared with the expected Mendelian 1:2:1 ratio. Statistical analyses were performed using the chi-squared test with two degrees of freedom (*$P < 0.05$, $\chi^2 > 5.991$). The data for P7 were adapted from Hoque et al (2024).

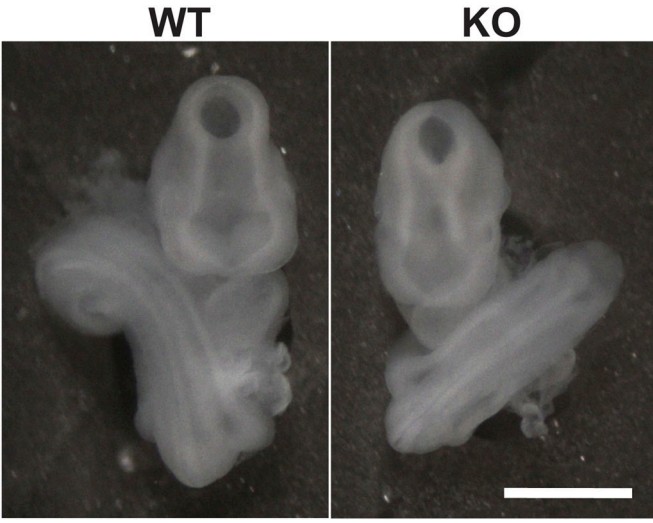

**WT**   **KO**

**Figure 1.  Left–right asymmetry defects in *ciBAR1*-KO embryos.**
Representative image of embryonic day 9.5 (E9.5) embryos. Note that the tail of the WT embryo is oriented such that the tail curves toward its right side (left image), whereas the tail of the KO embryo curves in the opposite direction (right image), suggestive of defective nodal cilia function. Scale bar, 1 mm.

The weights of both left and right *ciBAR1*-KO kidneys normalized to body weight were significantly larger than those of WT mice: 7.535 ± 0.1782 mg/g (SEM) for the WT left kidney versus 8.887 ± 0.3188 mg/g (SEM) for the *ciBAR1*-KO left kidney and 7.456 ± 0.1584 mg/g (SEM) for the WT right kidney versus 9.312 ± 0.5215 mg/g (SEM) for the *ciBAR1*-KO right kidney (Fig 3A). However, histological examination did not reveal major morphological differences or renal cysts commonly seen in ciliopathies (Fig 3B). The reason for the increased kidney weight in *ciBAR1*-KO mice is currently unknown.

## Ductal expansion in the pancreas of *ciBAR1*-KO mice

Perturbed acinar cell architecture, acinar-to-ductal metaplasia, and ductal dilation and expansion have been observed in the pancreas of ciliopathy mouse models (Lu et al, 1997; Cano et al, 2004, 2006; Augereau et al, 2016; Cyge et al, 2021) and human patients (van Asselt et al, 2013; Lodh et al, 2014). To further evaluate the increased duct-like structures in *ciBAR1*-KO pancreata (Fig 2C), pancreatic sections of adult WT and *ciBAR1*-KO mice were subjected to immunofluorescence (IF) staining with the ductal marker fluorescein-conjugated dolichos biflorus agglutinin (DBA) lectin and the zymogen granule marker rhodamine-labeled peanut agglutinin (PNA) lectin (Fig 4A). We found that there was a 1.7-fold increase in the number of ducts in *ciBAR1*-KO pancreata compared with WT samples (Fig 4B). In addition, quantitative analysis of the DBA lectin–positive ductal area and PNA lectin-positive acinar cell area in both groups revealed a 3.3-fold increase in the ductal area in *ciBAR1*-KO pancreata compared with WT tissue (Fig 4C). No significant changes in proliferation were detected using EdU cell proliferation assays (data not shown). These findings suggest that loss of ciBAR1 triggers acinar-to-ductal metaplasia, leading to ductal expansion in the pancreas.

## Characterization of the endocrine pancreas

PAS staining of adult pancreatic sections did not reveal any major differences in islet morphology between WT and *ciBAR1*-KO mice (Fig 5A). To further assess whether endocrine cell types are affected by loss of ciBAR1, we performed IF staining of adult pancreatic sections for glucagon (α-cell marker), insulin (β-cell marker), and somatostatin (δ-cell marker). As shown in Fig 5B, normal cell differentiation and architecture were observed in the pancreas of *ciBAR1*-KO mice.

Primary cilia in β-cells play critical roles in regulating glucose homeostasis, and their dysfunction is associated with development of diabetes (Hughes et al, 2020; Lee & Hughes, 2023). To examine if *ciBAR1*-KO mice show altered glucose metabolism, we performed the glucose tolerance test (GTT). After a 16-h fast and intraperitoneal injection of glucose, blood glucose levels in *ciBAR1*-KO and WT mice were monitored over 2 h (Fig 5C). Interestingly, *ciBAR1*-KO mice exhibited a moderate but consistent delay in blood glucose clearance compared with WT mice. In sum, although loss of ciBAR1 does not cause overt phenotypic changes in the endocrine pancreas, ciBAR1 influences glucose homeostasis likely through its role in ciliogenesis (see below).

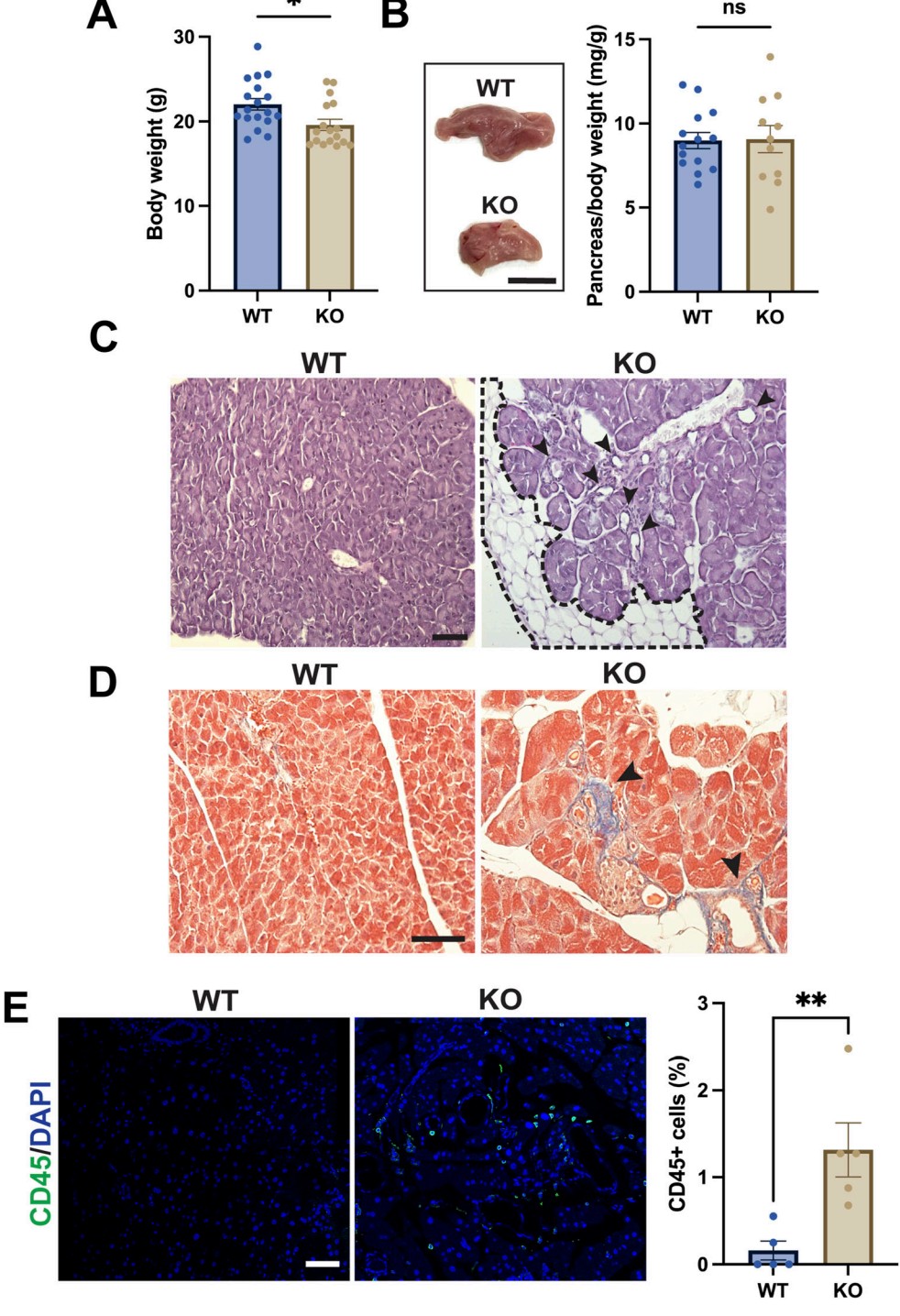

**Figure 2.  Exocrine pancreatic phenotypes of *ciBAR1*-KO mice.**
**(A)** Average body weights for adult WT (n = 18) and *ciBAR1*-KO (n = 17) mice. *P < 0.05.
**(B)** Representative images of pancreata from 2-mo-old mice. Scale bar, 0.5 cm. Adult WT (n = 14) and *ciBAR1*-KO (n = 11) pancreata were weighed and normalized to the body weight (mg/g). ns, not significant.
**(C)** Periodic acid–Schiff staining of pancreatic tissue from 2-mo-old mice shows extensive lipomatosis (outlined by the dashed line) and abundance of ducts (arrowheads) in the KO pancreas.
**(D)** Pancreatic sections were stained with Masson's trichrome to detect collagen deposition indicative of fibrosis (blue staining, arrowheads). Scale bar, 20 $\mu$m.
**(E)** Pancreatic sections from adult WT and *ciBAR1*-KO mice were immunostained for CD45 (leukocytes, green), and nuclei were visualized with DAPI (blue). The number of cells were counted in four to five representative 40X objective fields from each pancreas (n = 5 per genotype), and the percentage of CD45-positive (CD45[+]) cells was calculated. Scale bar, 50 $\mu$m. **P < 0.01. Adult mice (2–7 mo old) were used for the experiments unless indicated otherwise. For all graphs, data represent means ± SEM.

## Localization of ciBAR1 in the pancreas

We have previously shown that ciBAR1 localizes to the base of primary and multicilia in cultured human retinal pigment epithelial RPE1 cells and mouse tracheal epithelial cells, respectively (Li et al, 2016; Siller et al, 2017). To examine the subcellular localization of ciBAR1 in the adult pancreas, we performed IF staining using antibodies against the ciliary axonemal marker acetylated $\alpha$-tubulin (A-tub) and ciBAR1 in both WT and *ciBAR1*-KO pancreata (Fig 6). ciBAR1 was present at the base of primary cilia in the pancreatic ducts and islets of the WT pancreas but absent in the *ciBAR1*-KO pancreas, validating antibody specificity.

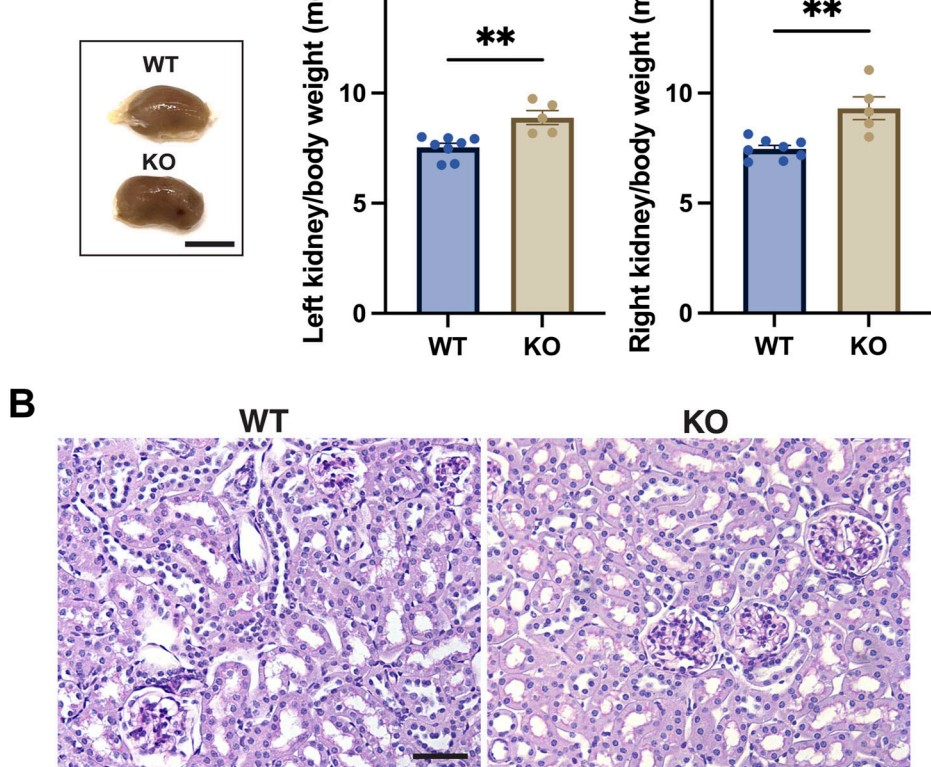

Figure 3.  No overt cystic phenotypes of *ciBAR1*-KO kidneys.
**(A)** Representative images of kidneys from 2-mo-old mice. Kidneys from adult WT (n = 8) and *ciBAR1*-KO (n = 5) mice were weighed and normalized to the body weight (mg/g). Data represent means ± SEM. **P < 0.01. Adult mice (2–7 mo old) were used for the experiments. **(B)** Representative images of PAS staining of WT and *ciBAR1*-KO kidneys from 2-mo-old mice. Scale bar, 20 μm.

## Defective primary cilia in the *ciBAR1*-KO pancreas

To investigate if loss of ciBAR1 influences the status of primary cilia in the pancreas, we performed IF staining for A-tub and the basal body marker γ-tubulin (G-tub) on pancreatic sections from adult WT and *ciBAR1*-KO mice (Fig 7A). Quantification of the number of ductal cilia revealed a 28.9% reduction in the *ciBAR1*-KO pancreas compared with WT samples. In addition, the average length of ductal cilia was decreased in the *ciBAR1*-KO pancreas: 4.895 ± 0.3918 μm (SEM) for WT cilia versus 2.303 ± 0.2235 μm (SEM) for *ciBAR1*-KO cilia. Similarly, in *ciBAR1*-KO islets, there was a 24.6% decrease in the ciliary count and a significant reduction in the ciliary length compared with WT samples: 4.748 ± 0.2095 μm (SEM) for WT cilia versus 2.784 ± 0.1601 μm (SEM) for *ciBAR1*-KO cilia. Similar results were obtained for IF staining using the ciliary membrane marker Arl13B and the basal body marker Centrin1 (Fig 7B). Furthermore, costaining of A-tub with the β-cell marker insulin revealed impaired ciliogenesis in β-cells in *ciBAR1*-KO mice (Fig 7C), consistent with their impaired glucose tolerance. In *ciBAR1*-KO ducts and islets, Cby1 was clearly detectable at the ciliary base (Fig S1). In contrast, ciBAR2 was undetectable at the ciliary base in both WT and *ciBAR1*-KO pancreata (Fig S1), suggesting that there is no compensation among family members.

## ciBAR1 is critical for ciliogenesis in MEFs

Primary cultures of MEFs have been widely used to study the formation and function of primary cilia. To further explore the role of ciBAR1 in primary cilia, we generated MEF cultures from WT and *ciBAR1*-KO E12.5 embryos. RT–PCR analysis showed that *ciBAR1*, but not its paralog *ciBAR2*, was expressed in WT MEFs (Fig 8A). Western blotting of WT and *ciBAR1*-KO MEF cell lysates confirmed complete loss of ciBAR1 in *ciBAR1*-KO MEFs (Fig 8B).

Next, we performed IF staining for Arl13B and G-tub to evaluate possible ciliary defects in *ciBAR1*-KO MEFs. We found that there was a 90.4% reduction in the number of MEFs with primary cilia in the absence of ciBAR1 (Fig 8C and D, top). In addition, the average cilium length of the remaining cilia was also reduced (Fig 8D, bottom). Interestingly, Arl13B-positive puncta were noticeable around the basal body in *ciBAR1*-KO MEFs (Fig 8C). We then examined the subcellular localization of ciBAR1, ciBAR2, and Cby1 (Fig 8E). In WT MEFs, ciBAR1 and Cby1 were detectable at the base of cilia. Cby1 was recruited to the ciliary base in *ciBAR1*-KO MEFs. ciBAR2 was undetectable at the ciliary base in both WT and *ciBAR1*-KO MEFs, suggesting that there is no compensatory up-regulation of ciBAR2 expression to mitigate loss of ciBAR1. Cby1 was also present at one of the G-tub positive centrioles, likely the mother centriole, in cycling *ciBAR1*-KO MEFs (Fig 8F).

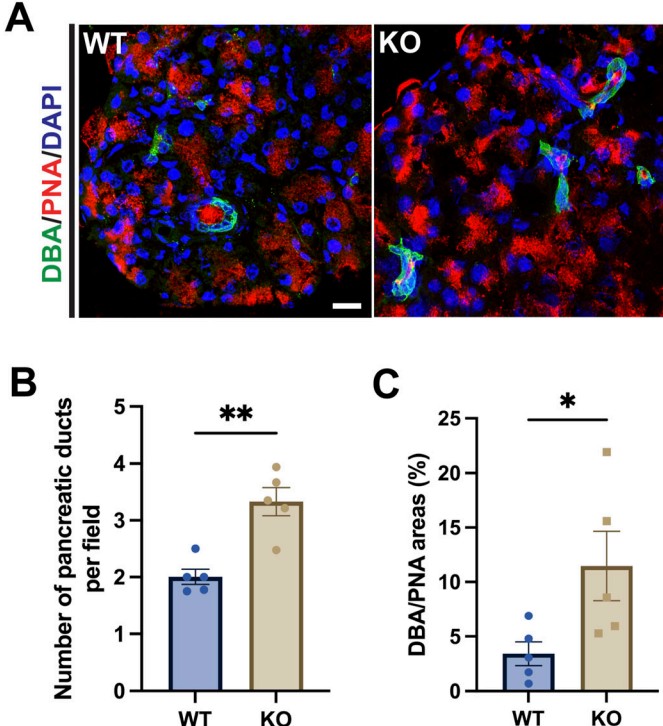

**Figure 4. Ductal expansion in *ciBAR1*-KO pancreata.**
**(A)** IF staining of WT and *ciBAR1*-KO pancreatic tissue sections from 2-mo-old mice with DBA lectin (ducts, green) and PNA lectin (zymogen granules, red). Nuclei were stained with DAPI (blue). Scale bar, 20 μm. **(B, C)** The average number of ducts (B) and the overall DBA-positive ductal area relative to the PNA-positive tissue area (C) were quantified in adult WT and *ciBAR1*-KO pancreata. For quantification, 15–20 representative 63X objective fields from each pancreas from adult mice (2–7 mo old) (n = 6 for WT and n = 7 for *ciBAR1* KO) were used for quantification. Data represent means ± SEM. *P < 0.05, **P < 0.01.

## Discussion

ciBAR1 belongs to the BAR superfamily, which regulates dynamic membrane processes such as membrane tubulation, invagination, and endocytosis (Salzer et al, 2017; Carman & Dominguez, 2018; Nishimura et al, 2018; Simunovic et al, 2019). We previously reported that ciBAR1 localizes to the base of cilia and plays an important role in ciliogenesis in cultured mammalian cells (Li et al, 2016). Consistent with its critical role in ciliogenesis, ciBAR1 has been previously implicated in non-syndromic postaxial polydactyly, a common feature of ciliopathies (Schrauwen et al, 2019; Umair et al, 2024). In addition, *ciBAR1*-KO mice exhibited altered digit morphologies, as evaluated by X-rays, such as polysyndactyly, metatarsal osteomas, and abnormalities on the deltoid tuberosity of the humerus (Schrauwen et al, 2019). However, its role in mammalian development and homeostasis remains poorly understood. Here, we demonstrated that ciBAR1 is important for ciliogenesis in mice, mediating proper left–right body asymmetry, exocrine pancreas development, and glucose metabolism. In the pancreas, *ciBAR1*-KO mice exhibited acinar cell degeneration, acinar-to-ductal metaplasia, disorganized tissue architecture, and inflammation and delayed glucose clearance. We found that in *ciBAR1*-KO mice, primary cilia present in pancreatic ducts and islets are significantly

reduced in number and length. Through IF staining, we showed that ciBAR1 localizes to the ciliary base in normal pancreatic ductal and islet cells. Similarly, *ciBAR1*-KO MEFs showed a robust decrease in the number and length of cilia. On the other hand, *ciBAR1*-KO mice did not display any overt renal pathologies. These results suggest that although *ciBAR1* is ubiquitously expressed in many different tissues like *Cby1*, its requirement is cell-type–specific, and loss of ciBAR1 might be compensated by other BAR domain–containing proteins in vivo.

The embryonic node is a transient cavity at the midline of the mouse embryo that is visible around E7.5 and then disappears by E9.0. The node contains two types of cilia: primary 9 + 0 motile cilia within the central region of the node that rotate in a clockwise direction at a speed of 600 rpm and immotile cilia at the periphery of the node (Hirokawa et al, 2009; Hamada, 2020). Left–right asymmetry is determined by the leftward extraembryonic fluid flow generated by the rotational movement and posterior tilt of motile cilia (Nonaka et al, 1998; Nonaka et al, 2002; Shinohara & Hamada, 2017). During the process of axial rotation, which occurs between the 8 and 12 somite stages, embryos turn in an anticlockwise fashion such that the head turns toward the right and the tail curves toward the left (Copp et al, 2023). Loss of nodal flow caused by the absence of motile nodal cilia or loss of cilia motility results in altered left–right patterning and laterality defects in both mice and humans (Huangfu & Anderson, 2005; Shiratori & Hamada, 2006; Fliegauf et al, 2007; Gorivodsky et al, 2009). Interestingly, single-cell RNA sequencing analysis of whole mouse embryos revealed that *ciBAR1* is expressed early in mouse development, and 20–100% of ciliated nodal cells express *ciBAR1* during the 0–11 somite stages (Fig S2) (Qiu et al, 2022, 2024), a critical window for determination of left–right asymmetry. In contrast, *ciBAR2* expression is undetectable in ciliated nodal cells at all stages of early mouse development. We did not observe any laterality defects such as situs inversus and heterotaxy in *ciBAR1*-KO postnatal and 2–7-mo-old mice, indicating that *ciBAR1*-KO embryos that exhibit abnormal left–right asymmetry die before birth. Although it seems likely that ciBAR1 is important for proper nodal cilia function, further experiments will be required to investigate potential ciliary defects in the *ciBAR1*-KO embryonic node.

Mutations that affect primary cilium formation and function often result in pancreatic and renal abnormalities such as cystic and fibrotic lesions (Frank et al, 2013; Lodh et al, 2014). Although *ciBAR1*-KO kidneys weighed significantly more than WT kidneys, *ciBAR1*-KO mice did not show any renal cysts or any other kidney-related pathological phenotype. However, *ciBAR1*-KO mice mirror many of the exocrine pancreatic defects—loss of acinar cell architecture, lipomatosis, ductal hyperplasia, inflammation, and collagen deposition—observed in KO mouse models of genes involved in cilia assembly and function, such as *Cby1*, *KIF3A*, *IFT88*, *PKD2*, and *inversin* (Cano et al, 2004, 2006; Cyge et al, 2021). These exocrine defects are also reported in human patients with pancreatic ductal adenocarcinoma and chronic pancreatitis (Cano et al, 2004). However, acinar cells, which comprise most of the cell population in the pancreas, notably do not possess cilia. If so, what is the mechanism underlying acinar cell degeneration caused by loss of ciBAR1? One possibility that has been previously proposed is that primary cilia present on pancreatic ducts act as

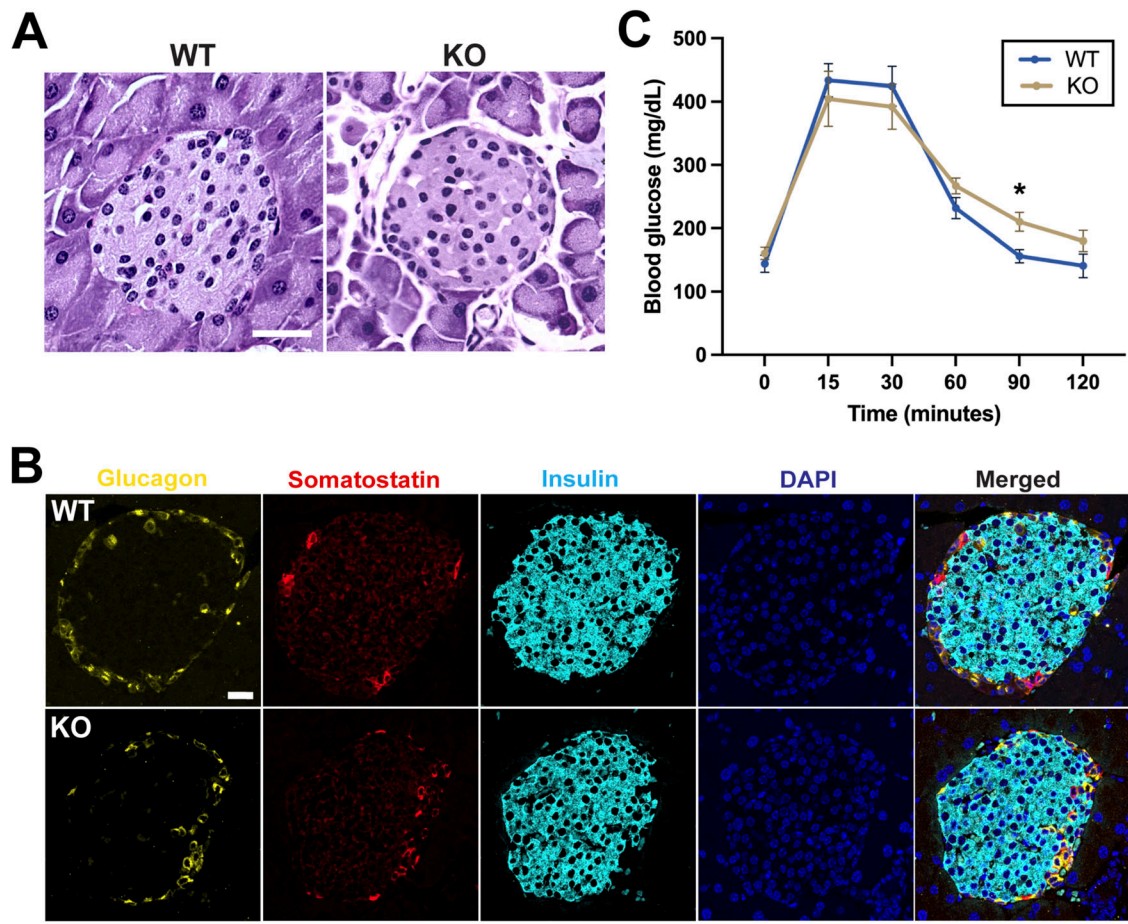

**Figure 5.    Evaluation of the endocrine pancreas.**
**(A)** PAS staining of pancreatic sections from 2-mo-old mice shows no significant difference in islet morphology between WT and *ciBAR1*-KO mice. Scale bar, 20 μm. **(B)** IF staining of pancreatic sections from 2-mo-old mice for glucagon (α-cells, yellow), somatostatin (δ-cells, red), and insulin (β-cells, cyan). Nuclei were stained with DAPI (blue). Scale bar, 20 μm. **(C)** Blood glucose tolerance tests were performed on adult WT (n = 5) and *ciBAR1*-KO (n = 5) mice (2–7 mo old) (left). Data represent means ± SEM. *$P$ < 0.05.

mechanosensory appendages that detect luminal fluid flow (Nauli & Zhou, 2004). Disruption of flow caused by defective cilia could result in increased pressure and the obstruction and subsequent dilation of pancreatic ducts, ultimately leading to acinar cell death (Cano et al, 2004, 2006; Cyge et al, 2021). Alternatively, ciBAR1 may play a cilia-independent role in acinar cells such as in sequential compound exocytosis, in which primary zymogen granules first fuse with the plasma membrane before subsequently fusing with secondary and tertiary zymogen granules (Vertii et al, 2015; Vitre et al, 2020; Cyge et al, 2021). We previously reported that *Cby1* is expressed in non-ciliated acinar cells and that loss of Cby1 leads to secretory defects and intracellular accumulation of abnormally interlinked zymogen granules, resulting in acinar cell apoptosis (Cyge et al, 2021). Interestingly, *Cby1*-KO zymogen granules did not show membrane fusion, which supports Cby1's role in ciliogenesis in facilitating the fusion of small pre-ciliary vesicles into the larger ciliary vesicle during basal body docking in complex with ciBAR proteins (Burke et al, 2014; Li et al, 2016). Like *Cby1*, *ciBAR1*, but not *ciBAR2*, is also expressed in acinar cells in both mice and humans (Fig S3). However, further investigation is required to understand

the molecular basis of acinar cell death in the *ciBAR1*-KO mouse model.

Aberrant glucose homeostasis is seen in a handful of cil-iopathies, such as BBS and Alstrom syndrome, which present as impaired glucose tolerance and/or insulin resistance (Marshall et al, 2005; Pietrzak-Nowacka et al, 2010; Gerdes et al, 2014; Lee & Hughes, 2023). Unlike the exocrine pancreas, defects in the en-docrine pancreas are less well defined in ciliopathy mouse models. *Cby1*- and *KIF3A*-KO mice show substantial exocrine defects but little or no significant change in islet architecture (Cano et al, 2004, 2006; Cyge et al, 2021). However, genetic ablation of the IFT com-ponent IFT88 and the ciligenesis transcription factor regulatory factor X3 (RFX3) in mice have been reported to show shortening or loss of cilia, impaired glucose tolerance, delayed insulin secretion, and changes in islet cell composition, implicating primary cilia in the regulation of signaling pathways that mediate endocrine cell differentiation and function (Ait-Lounis et al, 2007; Volta et al, 2019; Hughes et al, 2020; Lee & Hughes, 2023). It has recently been shown that β-cell cilia move in response to extracellular glucose and that ciliary motility genes are involved in modulating Ca²⁺ influx and

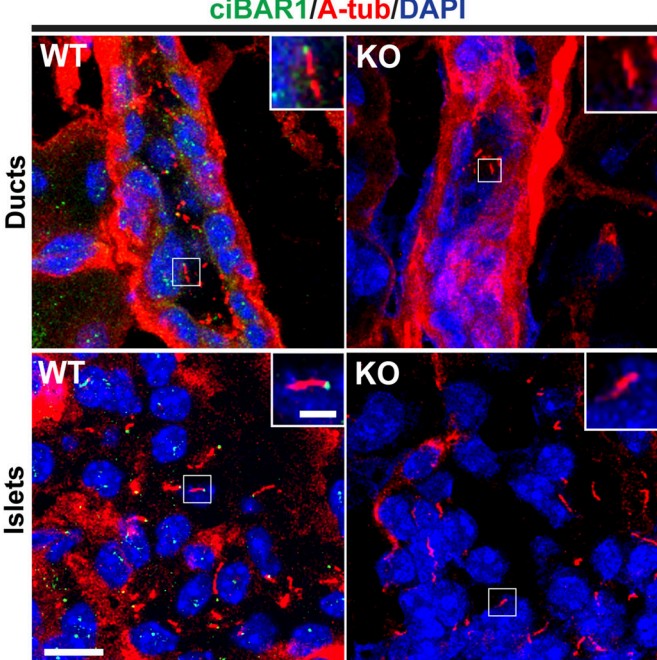

**Figure 6.  ciBAR1 localizes to the base of cilia in the pancreas.**
Pancreatic sections from 2-mo-old mice were immunostained for acetylated α-tubulin (A-tub) (ciliary axoneme, red) and ciBAR1 (green), and nuclei were detected with DAPI (blue). Scale bars, 10 and 1 μm (insets).

insulin secretion, suggesting that primary cilia motility may play a role in mediating glucose homeostasis and glucose-mediated paracrine signaling (Cho et al, 2022; Walker et al, 2023). Although *ciBAR1*-KO islets displayed normal cell morphology and no significant changes in cell mass or population, the absence of ciBAR1 resulted in reduced and shortened primary cilia in the islets and altered glucose tolerance, implicating ciBAR1 in the regulation of glucose metabolism. *ciBAR1* expression is detectable in islet cells (Fig S4). Although there was an overall trend toward delayed glucose clearance in *ciBAR1*-KO mice, a statistically significant difference was observed only at 90 min after glucose injection (Fig 5C). A similar glucose clearance pattern was reported in *BBS4*-KO mice (Gerdes et al, 2014). The heterogeneity of disease phenotypes in the endocrine pancreas across ciliopathy mouse models suggests that specific ciliopathy proteins may play crucial roles in regulating ciliary signaling pathways linked to the development and function of specific endocrine cell subtypes. The *ciBAR1*-KO mouse model thus enables us to take a closer look into the roles of primary cilia in both the endocrine and exocrine pancreas. Clearly, further work is warranted to elucidate the molecular role of ciBAR1 in controlling glucose homeostasis in the endocrine pancreas.

# Materials and Methods

### Mouse strain and ethics statement

All experimental procedures involving animals were approved by the Institutional Animal Care and Use Committee (IACUC) at Stony Brook University and conducted according to National Institute of Health (NIH) guidelines. Animals were housed in pathogen-free conditions on a 12-h light/dark cycle with free access to food and water within the Division of Laboratory Animal Resources at Stony Brook University.

The generation of *ciBAR1*-KO mice has been described previously (Hoque et al, 2024). Briefly, *ciBAR1* KO-first mice obtained from the Mary Lyon Centre (MRC) Harwell Institute were crossed with the C57BL/6-Tg (Zp3-cre) 93Knw/J mouse line (RRID:IMSR JAX:003651), which expresses Cre recombinase specifically in the oocyte. This resulted in removal of exon 2, which partially spans the BAR domain, leading to a frameshift mutation in germline *ciBAR1*-KO mice. *ciBAR1* heterozygous mice were then backcrossed over eight generations onto C57BL/6J (RRID:IMSR_JAX:000664) backgrounds to mitigate genetic drift and maintain phenotypic consistency. Adult mice (2–7 mo old) were used for the experiments unless specified otherwise. To assess the ages of embryonic day (E) 9.5 mice, the presence of a vaginal plug around noon in pregnant *ciBAR1* heterozygous mice was used to identify pup age E0.5.

Genotyping was performed via PCR using genomic DNA obtained from toe and tail clips as described previously (Hoque et al, 2024). The following primers were used: *ciBAR1* WT allele, 5′-CTGAAGGAGGATGCTTGGTGTTCCC-3′ and 5′-GAAGGGCAACATTCCCAACTCTTCC-3′ (650 bp); and *ciBAR1* KO allele, 5′-GCTACCATTAC-CAGTTGGTCTGGTGTC-3′ and 5′-TGGTCACTCAAAGCAAACACACAGC-3′ (586 bp).

### Preparation of MEFs

MEFs were prepared as described (Siller et al, 2017). In brief, E12.5 embryos were isolated from pregnant females after $CO_2$ asphyxiation. Embryonic tissue, excluding head, liver, and heart, was transferred to a 1.5-ml tube with 250–500 μl of 0.05% Trypsin–EDTA in PBS, finely minced with spring scissors, and incubated at 37°C for 30 min. The suspension was loaded onto a 3-ml syringe and passed through with a 20-gauge needle six times before cells were plated out in a six-well dish containing DMEM + 10% FBS + 100 U penicillin/streptomycin (P/S). MEFs were grown at 37°C, dissociated, and plated out on glass coverslips in a 12-well dish containing DMEM + 10% FBS + 100 U P/S. Once the cells reached confluency, primary ciliogenesis was induced: cells were washed once with PBS and incubated in serum-free media (DMEM + 100 U P/S) for 48 h. Genotyping was performed using genomic DNA obtained from embryonic heads.

### RT–PCR

Total RNA was isolated from MEFs using an RNeasy Mini Kit (QIAGEN) according to the manufacturer's instructions. Single-stranded cDNA was synthesized using a high-capacity cDNA reverse transcription kit (Applied Biosystems). PCR primers used were as follows: *ciBAR1*, 5′-AACAACTGCAGGATGCTGTC-3′ and 5′-AGTCTTTCAACCTCTGCCTG-3′ (232 bp); *ciBAR2*, 5′-ACACGGTGACAAATGCTGAG-3′ and 5′-TTCTGCCTCAGTTTCTCCAG-3′ (355 bp); and 18S ribosomal RNA (*r18S*), 5′-CTCAACACGGGAAACCTCAC-3′ and 5′-CGCTCCACCAACTAA-GAACG-3′ (110 bp).

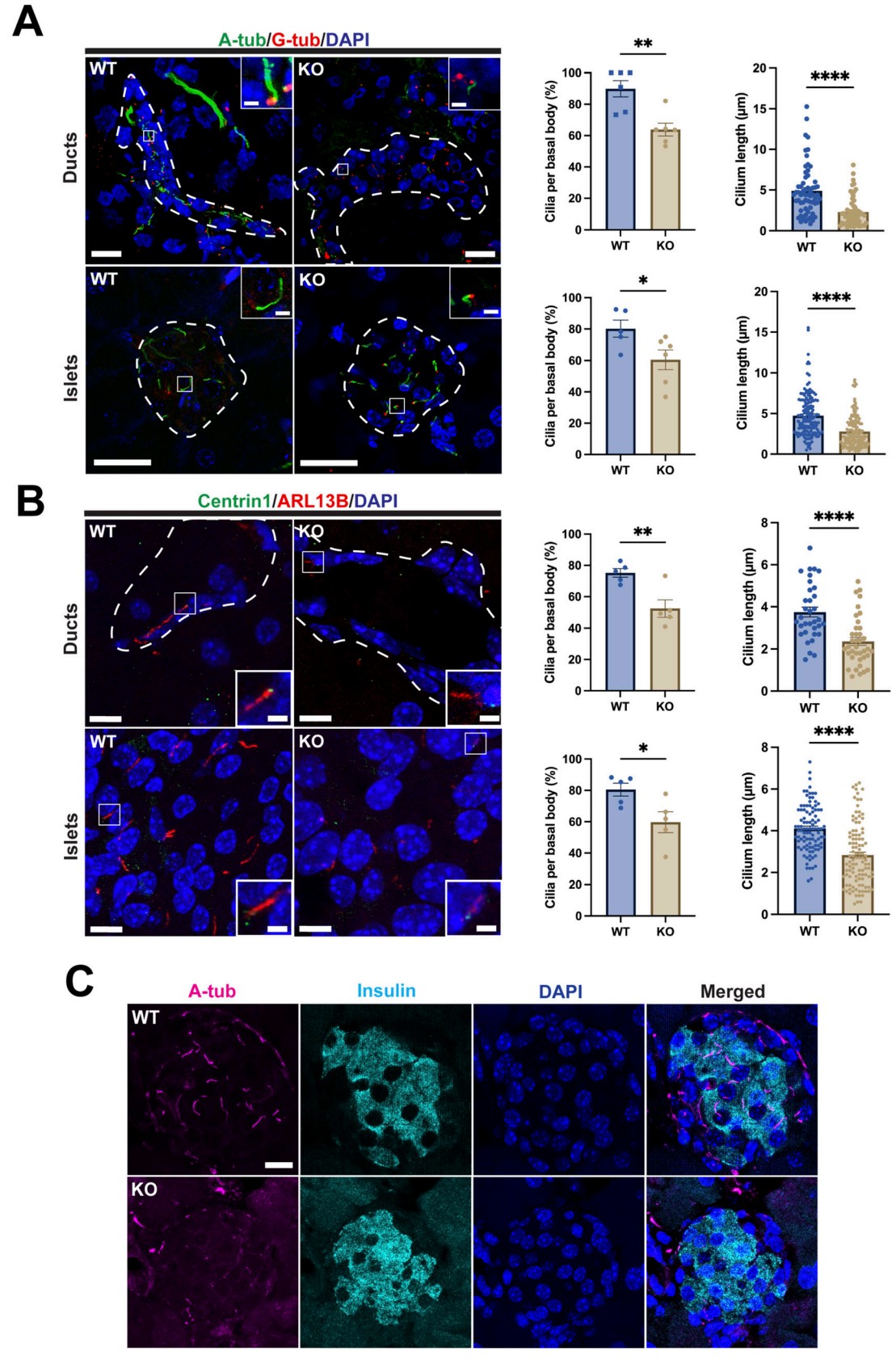

## Western blotting

Western blotting was performed as described previously (Hoque et al, 2024). In brief, MEFs were lysed in ice-cold RIPA buffer (50 mM Tris–HCl, pH 8.0, 150 mM NaCl, 5 mM EDTA, 1% NP-40, 0.5% sodium deoxycholate, and 0.1% SDS) containing protease inhibitor cocktail (Roche) and incubated for 20 min on ice. Cell lysates were sonicated and cleared by centrifugation at 13,000$g$ for 30 min at 4°C, followed by separation on SDS–PAGE. The primary antibodies used were rabbit ciBAR1 (1:500, 24803-1-AP, RRID:AB_2879735; Proteintech), and mouse GAPDH (1:10,000, H86504M; Biodesign International) antibodies. HRP-conjugated secondary antibodies were purchased from Jackson ImmunoResearch Laboratories. Protein bands were visualized using chemiluminescence (SuperSignal West Pico PLUS Chemiluminescent Substrate; Pierce) with x-ray films, and the developed films were then scanned.

## Histological analysis

Pancreata or kidneys dissected from adult mice were embedded in paraffin blocks, after 4% PFA fixation overnight at 4°C. 5-$\mu$m sections were cut using a Leica RM2125 microtome, deparaffinized, rehydrated, stained with hematoxylin and eosin (Poly Scientific R&D Corp) or a PAS kit (Sigma-Aldrich), and mounted with Permount mounting medium (Thermo Fisher Scientific). Masson's Trichrome staining on paraffin-embedded tissues was performed by the Research Histology core at Stony Brook University.

## IF staining

MEFs on glass coverslips were fixed in ice-cold methanol-acetone mixture (1:1) for 10 min, washed three times with PBS for 10 min each at 4°C, and blocked for 1 h at room temperature with 5% goat serum in antibody diluent buffer (2% BSA and 0.2% Triton X-100 in PBS). Samples were then incubated with primary antibody in antibody diluent buffer overnight at 4°C, followed by three washes with PBS for 5 min each. Next, samples were incubated with secondary antibodies in antibody diluent buffer for 1 h at room temperature. Subsequently, samples were washed three times with PBS for 5 min each. Finally, DAPI counterstain was performed for 2 min at room temperature, followed by three washes with PBS for 5 min each. Samples were then mounted with Fluoromount-G (Southern Biotech).

5-$\mu$m paraffin sections were deparaffinized, rehydrated, and washed with deionized water. Heat-induced epitope retrieval was performed by autoclaving tissue samples in a citrate-based antigen unmasking solution (Vector Laboratories) for 17.5 min. Samples were cooled to room temperature for 30 min before permeabilization with 0.5% Triton X-100 in PBS for 10 min. For immunostaining of frozen sections, pancreata were placed in optimal cutting temperature compound in a plastic embedding mold and immediately snap-frozen in isopentane (2-methylbutane) chilled in liquid nitrogen. Blocks were stored at –80°C until use. 10-$\mu$m sections were cut using a Leica CM1950 cryostat and fixed with ice-cold methanol-acetone mixture (1:1) for 10 min. Tissue samples were blocked with 5% goat or donkey serum in dilution buffer (2% BSA in PBS) for 1 h at room temperature and then washed three times in wash solution (0.05% Tween-20 in PBS) for 5 min each. Samples were incubated in primary antibodies diluted in dilution buffer overnight at 4°C. On the following day, samples were given three 5-min washes with wash solution before incubation in secondary antibodies in dilution buffer for 1 h at room temperature. Samples were washed three times for 5 min each with wash solution and then incubated with DAPI for 5 min at room temperature. After two quick rinses with PBS, samples were mounted with Fluoromount-G (Southern Biotech).

The primary and secondary antibodies and lectins used for IF staining are listed in Table S1.

## Image acquisition and quantification

PAS- and hematoxylin and eosin–stained tissues were imaged using a Leica DFC7000T camera mounted on a Leica DMI6000B microscope using the following objective lenses: 20X/0.50 NA, 40X/0.75 NA, or 63X/1.25 NA oil. Fluorescent images were obtained on a Leica SP8X with a HCPL APO 100X/1.4 NA objective and a Zeiss LSM 980 with Airyscan 2 with Plan-Apochromat using 40X/1.4 NA water, 63X/1.40 NA oil, or 100X/1.4 NA oil objectives. All images were processed with the Leica Application Suite X or Zeiss Zen Blue programs. Further image processing was performed via ImageJ/FIJI, Adobe Photoshop, CellProfiler, and Adobe Illustrator.

Quantification of CD45$^+$ immune cells was systematically performed on IF images using a CellProfiler-based analytical pipeline. The percentage of CD45$^+$ cells was assessed by dividing the number of CD45$^+$ cells by the total number of DAPI+ nuclei per field. Ductal numbers and areas in the pancreas were also obtained using a CellProfiler-based pipeline. Images of DBA- and PNA-lectin–stained pancreatic sections were thresholded using a global threshold strategy and a two-class Otsu segmentation on each image. Each DBA+ ductal region was identified manually.

**Figure 7. Decreased cilia number and length in *ciBAR1*-KO pancreatic ducts and islets.**
**(A)** Pancreatic sections from adult mice (2–7 mo old) were immunostained for A-tub (green) and G-tub (basal bodies, red) to identify primary cilia in pancreatic ducts and islets (encircled by the dashed line). Nuclei were stained with DAPI (blue). Representative images from 2-mo-old mice are shown. Scale bars, 10 and 1 $\mu$m (insets). Quantification of the number and length of primary cilia is shown on the right. For the graph of cilium numbers plotted as the percentage of cilia per basal body, each point represents the average number of primary cilia from four to five representative 40X objective fields from each pancreas (n = 5 for WT and n = 6 for *ciBAR1* KO). For the graph of cilium lengths, each point represents the length of individual primary cilia. A total of 62 WT and 61 *ciBAR1*-KO ductal cilia per mouse and 157 WT and 141 ciBAR1-KO islet cilia per mouse were quantified. Data represent means ± SEM. *P < 0.05, **P < 0.01, ****P < 0.0001. **(B)** Pancreatic sections from adult mice (2–7 mo old) were immunostained for ARL13B (ciliary membrane, green) and Centrin1 (basal bodies, red) (n = 5 for WT and n = 5 for *ciBAR1* KO). Representative images from 2-mo-old mice are shown. Scale bars, 10 and 1 $\mu$m (insets). Quantification of the number and length of primary cilia is shown on the right. A total of 34 WT and 28 *ciBAR1*-KO ductal cilia per mouse and 100 WT and 107 *ciBAR1*-KO islet cilia per mouse were quantified. Data represent means ± SEM. *P < 0.05, **P < 0.01, ****P < 0.0001. **(C)** IF staining of pancreatic sections from 2-mo-old mice for A-tub (magenta) and insulin (green). Nuclei were stained with DAPI (blue). Scale bar, 20 $\mu$m.

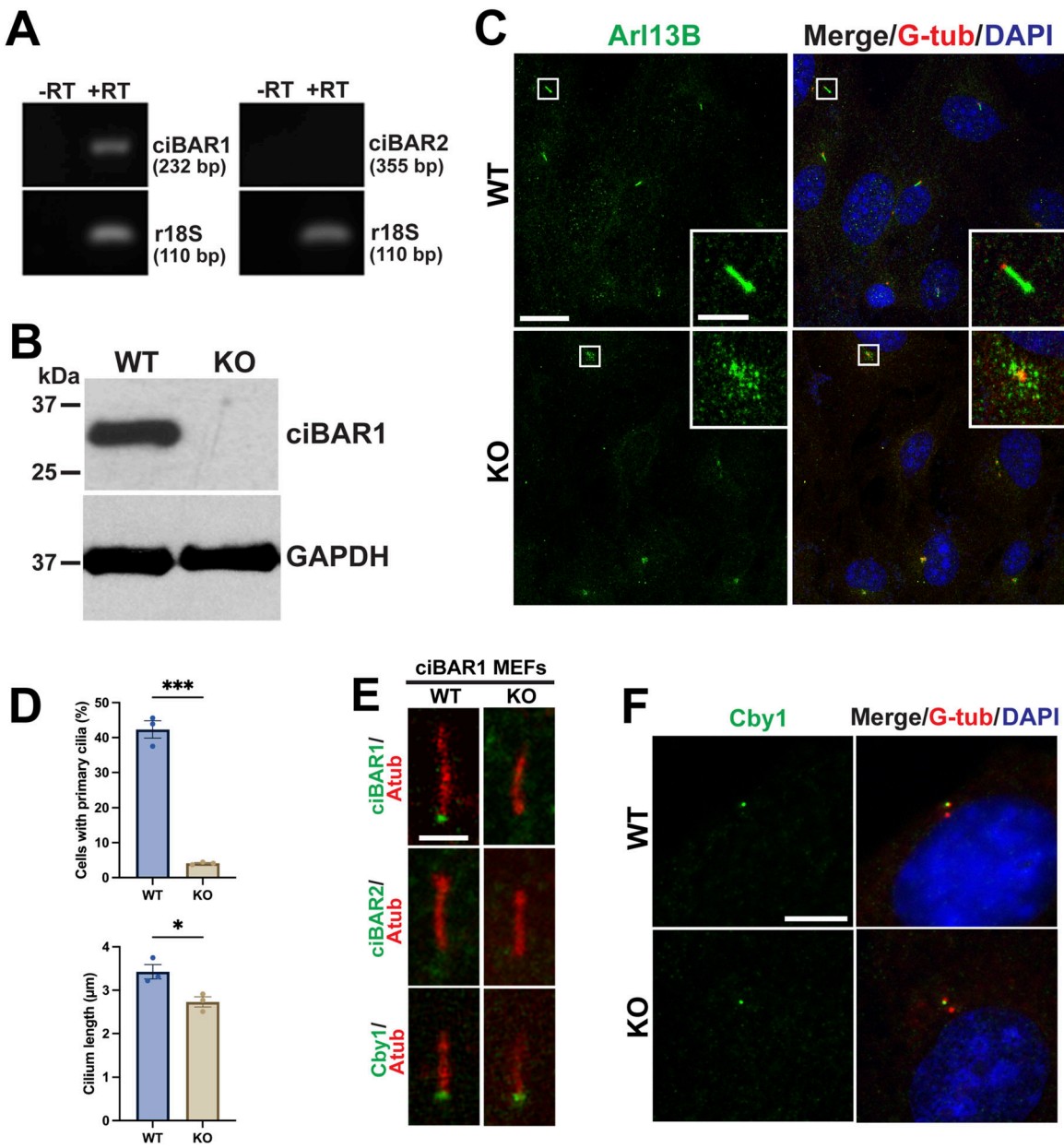

**Figure 8. ciBAR1 plays a crucial role in ciliogenesis in MEFs.**
**(A)** RT–PCR of MEFs for ciBAR1 and 2. 18S ribosomal RNA was used as an internal control. **(B)** Cell lysates from WT or *ciBAR1*-KO MEFs were analyzed by Western blotting for ciBAR1. GAPDH was used as a loading control. **(C)** WT and *ciBAR1*-KO MEFs were immunostained for Arl13B (green) and G-tub (red) after serum starvation for 48 h. Nuclei were detected with DAPI (blue). The squared areas are magnified in the insets. Scale bars, 20 and 5 μm (insets). **(C, D)** Quantification of the number of cells with primary cilia (top) and primary cilium length (bottom) in WT and KO MEFs in (C). The number of cells with primary cilia was scored from three independent experiments (n = 427 cells for WT and n = 470 cells for KO MEFs). Similarly, the cilium length was quantified from three independent experiments (n = 484 cilia for WT and n = 273 cilia for KO MEFs). Data represent means ± SEM. *P < 0.05. **(E)** WT and *ciBAR1*-KO MEFs were immunostained for ciBAR1, ciBAR2, or Cby1 (green) and A-tub (red) as indicated. Scale bar, 2 μm. **(F)** Cycling MEFs were immunostained for Cby1 (green) and G-tub (centrioles, red). Nuclei were visualized with DAPI. Scale bar, 5 μm.

The number of ducts per field was automatically assessed by the pipeline and was averaged for each image. The ratio of the ductal area relative to the exocrine area was calculated by dividing the DBA+ area by the PNA+ area. Measurement of ciliary length was manually performed using the segmented line selection tool in ImageJ/FIJI.

**Intraperitoneal (IP) GTT**

2–3-mo-old male mice were fasted for ~16 h with free access to water. Baseline measurements of blood glucose levels were taken with a glucometer before mice were given an intraperitoneal glucose injection of 50% dextrose sterile preservative-free solution

(VET ONE) (2 g/kg of body weight). Blood glucose levels were measured at 15, 30, 60, 90, and 120 min after injection.

### Statistical analysis

Statistical analyses were performed using unpaired, two-tailed $t$ tests. Quantification and statistical analysis of GTT were conducted as described (Anhê et al, 2022). Statistical significance in blood glucose levels between the two groups at specific time points was determined using a two-way, repeated-measures ANOVA with Tukey's post-test. $P < 0.05$ was considered to be significant. Asterisks were used to indicate $P$-values as follows: $*P < 0.05$; $**P < 0.01$; $***P < 0.001$; and ns, not significant. GraphPad Prism was used for all statistical analyses and graphical representations.

## Supplementary Information

## Acknowledgements

We would like to thank the Research Histology and Central Microscopy Center Core Facilities at Stony Brook University Renaissance School of Medicine for assistance with histological preparations and confocal imaging, respectively. This work was supported by grants from National Heart, Lung, and Blood Institute (R01HL139643 to K-I Takemaru and F31HL168828 to EN Kim) and National Institute of Diabetes and Digestive Kidney Diseases (R01DK123641 to K-I Takemaru).

### Author Contributions

EN Kim: conceptualization, data curation, formal analysis, funding acquisition, investigation, methodology, and writing—original draft.
F-Q Li: conceptualization, data curation, formal analysis, investigation, methodology, and writing—review and editing.
K-I Takemaru: conceptualization, formal analysis, supervision, funding acquisition, investigation, methodology, project administration, and writing—review and editing.

### Conflict of Interest Statement

The authors declare that they have no conflict of interest.

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
