## [Reviewer comments · Life Science Alliance]

Life Science Alliance

ciBAR1 loss in mice causes laterality defects, pancreatic degeneration and altered glucose tolerance

Eunice Kim, Feng-Qian Li, and Ken-Ichi Takemaru

DOI: <https://doi.org/10.26508/lsa.202402916>

Corresponding author(s): Ken-Ichi Takemaru, Stony Brook University

Review Timeline:

Submission Date:	2024-06-30
Editorial Decision:	2024-08-23
Revision Received:	2024-11-20
Editorial Decision:	2024-11-22
Revision Received:	2024-11-25
Accepted:	2024-11-25

Transaction Report:

August 23, 2024

Re: Life Science Alliance manuscript #LSA-2024-02916

Dr. Ken-Ichi Takemaru
Stony Brook University
Dept. of Pharmacological Sciences BST7-182, Nicolls Rd.
Stony Brook, NY 11794-8651

Dear Dr. Takemaru,

Thank you for submitting your manuscript entitled "Loss of the ciliary BAR-domain protein ciBAR1 causes laterality defects, pancreatic exocrine degeneration, and altered glucose tolerance in mice" to Life Science Alliance. The manuscript was assessed by expert reviewers, whose comments are appended to this letter. We invite you to submit a revised manuscript addressing the Reviewer comments.

Thank you for this interesting contribution to Life Science Alliance. We are looking forward to receiving your revised manuscript.

Sincerely,

B. MANUSCRIPT ORGANIZATION AND FORMATTING:

Reviewer #1 (Comments to the Authors (Required)):

This study by Kim et al characterizes the ciBAR1 KO mouse, continuing on the group's previous work showing that ciBAR1 localizes to the cilia base and regulates ciliogenesis. The authors do a good job assessing the ciliopathy-related phenotypes of this mouse, highlighting pancreatic changes that result in glucose dysregulation, mirroring that seen in human ciliopathy patients. The pancreatic evaluations are well-chosen and carefully done, and the data is compelling. Overall I find this to be a well-executed study and of topical importance to the cilia and metabolism research fields.

Suggestions for improvement:

- Why pancreas? The discussion mentions scSeq data on ciBAR1 in mouse nodal cells; what about other organs? i.e. is the prominence of pancreatic phenotypes explained by selective expression of ciBAR1 in pancreatic tissues, and if so, which endocrine/exocrine cell types? This can be mined from existing mouse/human pancreas scSeq datasets. Would also recommend adding to Fig 5B cilia staining by acetylated alpha tubulin to test for cell type-specific changes in cilia expression, particularly in beta cells co-labeled with insulin.
- To round out the mechanism, what happens to Cby1/Cby3 when ciBAR is knocked out, are there compensatory changes in their expression or complexing with ciBAR2, for example?
- At least in human pancreas, acinar cells actually have detectable cilia gene expression by scSeq. It may be of interest to re-examine acinar cilia and centrosome expression and function in your ciBAR1 KO model given its severe acinar phenotype.

Referee Cross-Comments:

Agree with Reviewer 2 suggestions for additional experimental details, clarifications, and discussions of study limitations.

Reviewer #2 (Comments to the Authors (Required)):

In this manuscript, the authors describe in more detail the phenotype of the ciBAR1 (FAM92A) KO mice previously used to demonstrate the role of ciBAR1 during spermatogenesis. The authors show that ciBAR1 KO embryos exhibit a significant proportion (25%) of inverted rotation at E9.5. They propose that this phenotype is responsible for distorted Mendelian frequencies at birth, as none of the newborn mice show laterality defects. In addition, the authors describe two other organ defects: while the kidney seems to show no major histological defects except for an increase in weight, the exocrine pancreas shows pancreatic lesions with acinar degeneration and ductal dilatation, suggesting acinar-to-ductal metaplasia. The endocrine pancreas shows no obvious histological defects, but the ciBAR1 KO mouse has a slight delay in glucose blood clearance associated with a reduced number and size of pancreatic cilia. Furthermore, the authors show that the same reduction in cilia number can be observed in MEF derived from the ciBAR1 KO mouse.

This study is limited to the description of the KO mouse, but it will be helpful to the community working on the pathophysiological mechanisms of ciliopathies, as it will draw attention to the affected organs in this new ciliopathy model.

I have only a few suggestions for the authors before publication:

-It is not clear at what exact age the phenotypic analysis of the kidney and pancreas was performed. Could the authors add this information to the figure legend? What is the oldest age observed for these phenotypic analyses? Is there a difference between young and old mice, does the observed phenotype worsen? For the kidney, could any phenotype appear much later than what was observed?

-In the pancreas, the authors use Ac-Tub and Gamma-Tub to detect the cilium and centriole. Their images show significant non-"centriolar or ciliary" staining, which based on these images probably makes quantification of cilia difficult. Could the authors use other centriolar (centrin?) or ciliary (Arl13b, other?) staining to improve quantification accuracy and confirm their quantifications?

-The difference in glucose clearance between control and KO mice is only significant 90 min after glucose injection: is this correct? Please state clearly. Is there any other ciliary mouse model with such small differences?

-In MEF the authors only quantified the reduction in cilia number, is there any reduction in ciliary length too?

-The authors suggest that the altered rotation of the embryo could be directly due to ciliary defects in the embryonic node. This is indeed the most likely hypothesis, but there are no data directly showing ciliary defects in the node. The authors should mention this limitation.

-The same model has also been described in another study, independent of these authors, showing that mice have digit and limb phenotypes. The authors should at least cite this other study in the discussion.

-Human patients with defects in *ciBAR1* have been described. Again, could the authors comment on the phenotype of the mice compared to what is known about the patients?

Response to the reviewers

We thank the reviewers for their critical assessment of our work and for the opportunity to revise our submitted manuscript, "ciBAR1 loss in mice causes laterality defects, pancreatic exocrine degeneration and altered glucose tolerance." We address the reviewers' comments below.

Reviewer 1

1: Why pancreas? The discussion mentions scSeq data on ciBAR1 in mouse nodal cells; what about other organs? i.e. is the prominence of pancreatic phenotypes explained by selective expression of ciBAR1 in pancreatic tissues, and if so, which endocrine/exocrine cell types? This can be mined from existing mouse/human pancreas scSeq datasets. Would also recommend adding to Fig 5B cilia staining by acetylated alpha tubulin to test for cell type-specific changes in cilia expression, particularly in beta cells co-labeled with insulin.

Reply: Like Cby1, ciBAR1 is ubiquitously expressed in various tissues and cell types in mice and humans. In this manuscript, we focused on the organs and cell types that are sensitive to ciliary dysfunction and have not extensively examined other organs. However, upon necropsy, we did not find any overt abnormalities in other organs. It is worth noting that the exocrine pancreas is one of the most severity affected organs in Cby1-KO mice [1]. These findings suggest that ciBAR1 and Cby1 play major roles in ciliogenesis in some, but not all tissues and/or compensatory mechanisms exist *in vivo*. Analysis of existing scRNA-seq revealed that ciBAR1 is highly expressed in endocrine islet cells (alpha, beta, and delta cells) in both mice and humans (**new Fig. S4**).

We have now included immunofluorescence staining of beta-cells for insulin and the ciliary marker acetylated alpha-tubulin (A-tub) to test for beta-cell-specific changes in cilia expression (**new Fig. 7C**).

2: To round out the mechanism, what happens to Cby1/Cby3 when ciBAR is knocked out, are there compensatory changes in their expression or complexing with ciBAR2, for example?

Reply: To test for possible compensatory changes in expression of ciBAR2 and/or Cby1, we performed immunofluorescence staining for ciBAR2 and Cby1 with A-tub (**new Fig. S1**). Cby1 was detectable at the base of both ductal and islet cilia in both WT and ciBAR1-KO pancreata. ciBAR2 expression is mostly restricted to multiciliated cells in the airway, ependyma, and female and male reproductive systems, and we could not detect ciBAR2 protein in both WT and ciBAR1-KO pancreata by immunofluorescence staining. This is similar to what we have observed in WT and ciBAR1-KO MEFs (**Fig. 8E**). Both Cby2 and Cby3 are expressed exclusively in the testis and not in the pancreas.

3: At least in human pancreas, acinar cells actually have detectable cilia gene expression by scSeq. It may be of interest to re-examine acinar cilia and centrosome expression and function in your ciBAR1 KO model given its severe acinar phenotype.

Reply: This is an excellent point which we have also considered. However, acinar cells do not express cilia [2-4], and it seems more likely that these cilia-related genes play non-ciliary roles within the cell.

Reviewer 2

1: It is not clear at what exact age the phenotypic analysis of the kidney and pancreas was performed. Could the authors add this information to the figure legend? What is the oldest age observed for these phenotypic analyses? Is there a difference between young and old mice, does the observed phenotype worsen? For the kidney, could any phenotype appear much later than what was observed?

Reply: The figure legend has been updated to include the age information. The phenotypic analysis was performed on 2- to 7-month-old mice, and the observed pancreatic phenotypes did not appear to worsen between young and old mice. It is possible that renal disease phenotypes may appear in mice at ages later than the oldest mouse used in our study (7 months old).

2: In the pancreas, the authors use Ac-Tub and Gamma-Tub to detect the cilium and centriole. Their images show significant non-"centriolar or ciliary" staining, which based on these images probably makes quantification of cilia difficult. Could the authors use other centriolar (centrin?) or ciliary (Arl13b, other?) staining to improve quantification accuracy and confirm their quantifications?

Reply: We used the centriolar marker Centrin1 and ciliary membrane marker ARL13B to confirm our quantifications of ciliary length and number (**new Figure 7B**).

3: The difference in glucose clearance between control and KO mice is only significant 90 min after glucose injection: is this correct? Please state clearly. Is there any other ciliary mouse model with such small differences?

Reply: Yes, the difference in glucose clearance between control and KO mice is only significant 90 minutes after glucose injection. Intriguingly, this is also seen in *Bardet-Biedl-Syndrome 4 (Bbs4)* mutant mice, which shows impaired glucose clearance 80 minutes after glucose injection [5]. We have clearly stated this in the last paragraph of the Discussion as "Although there was an overall trend towards delayed glucose clearance in ciBAR1-KO mice, a statistically significant difference was observed only at 90 minutes after glucose injection (Fig 5C). A similar glucose clearance pattern was reported in BBS4-KO mice."

4: In MEF the authors only quantified the reduction in cilia number, is there any reduction in ciliary length too?

Reply: We have revisited the MEF quantification and found that not only do ciBAR1-KO MEFs have a reduced number of cilia, but the remaining cilia also show a significant reduction in ciliary length (**new Fig. 8D**).

5: The authors suggest that the altered rotation of the embryo could be directly due to ciliary defects in the embryonic node. This is indeed the most likely hypothesis, but there are no data directly showing ciliary defects in the node. The authors should mention this limitation.

Reply: This is an excellent point that we neglected to mention. We have updated the Discussion section to include this limitation at the end of the 2nd paragraph as “While it seems likely that ciBAR1 is important for proper nodal cilia function, further experiments will be required to investigate potential ciliary defects in the ciBAR1-KO embryonic node.”

6: The same model has also been described in another study, independent of these authors, showing that mice have digit and limb phenotypes. The authors should at least cite this other study in the discussion. Human patients with defects in ciBAR1 have been described. Again, could the authors comment on the phenotype of the mice compared to what is known about the patients?

Reply: Thank you for the insightful comments. We have revised to include the previous studies [6, 7] in the first paragraph of the Discussion section as “Consistent with its critical role in ciliogenesis, ciBAR1 has been previously implicated in non-syndromic postaxial polydactyly, a common feature of ciliopathies. In addition, ciBAR1-KO mice exhibited altered digit morphologies, as evaluated by X-rays, such as polysyndactyly, metatarsal osteomas, and abnormalities on the deltoid tuberosity of the humerus.”

References:

1. Cyge B, Voronina V, Hoque M, Kim EN, Hall J, Bailey-Lundberg JM, et al. Loss of the ciliary protein Chibby1 in mice leads to exocrine pancreatic degeneration and pancreatitis. *Sci Rep.* 2021;11(1):17220. Epub 2021/08/28. doi: 10.1038/s41598-021-96597-w. PubMed PMID: 34446743; PubMed Central PMCID: PMC8390639.
2. Cano DA, Murcia NS, Pazour GJ, Hebrok M. Orpk mouse model of polycystic kidney disease reveals essential role of primary cilia in pancreatic tissue organization. *Development.* 2004;131(14):3457-67. doi: 10.1242/dev.01189. PubMed PMID: 15226261.
3. Ait-Lounis A, Baas D, Barras E, Benadiba C, Charollais A, Nlend Nlend R, et al. Novel function of the ciliogenic transcription factor RFX3 in development of the endocrine pancreas. *Diabetes.* 2007;56(4):950-9. Epub 20070117. doi: 10.2337/db06-1187. PubMed PMID: 17229940.

4. Lodh S, O'Hare EA, Zaghloul NA. Primary cilia in pancreatic development and disease. *Birth Defects Res C Embryo Today*. 2014;102(2):139-58. Epub 20140526. doi: 10.1002/bdrc.21063. PubMed PMID: 24864023; PubMed Central PMCID: PMC4213238.
5. Gerdes JM, Christou-Savina S, Xiong Y, Moede T, Moruzzi N, Karlsson-Edlund P, et al. Ciliary dysfunction impairs beta-cell insulin secretion and promotes development of type 2 diabetes in rodents. *Nat Commun*. 2014;5:5308. Epub 20141106. doi: 10.1038/ncomms6308. PubMed PMID: 25374274.
6. Schrauwen I, Giese AP, Aziz A, Lafont DT, Chakchouk I, Santos-Cortez RLP, et al. FAM92A Underlies Nonsyndromic Postaxial Polydactyly in Humans and an Abnormal Limb and Digit Skeletal Phenotype in Mice. *J Bone Miner Res*. 2019;34(2):375-86. Epub 20181105. doi: 10.1002/jbmr.3594. PubMed PMID: 30395363; PubMed Central PMCID: PMC6489482.
7. Umair M, Ahmed Z, Shaker B, Bilal M, Al Abdulrahman A, Khan H, et al. A novel homozygous FAM92A gene (CIBAR1) variant further confirms its association with non-syndromic postaxial polydactyly type A9 (PAPA9). *Clin Genet*. 2024;106(4):488-93. Epub 20240610. doi: 10.1111/cge.14572. PubMed PMID: 38853702.

November 22, 2024

RE: Life Science Alliance Manuscript #LSA-2024-02916R

Dr. Ken-Ichi Takemaru
Stony Brook University
Dept. of Pharmacological Sciences BST7-182, Nicolls Rd.
Stony Brook, NY 11794-8651

Dear Dr. Takemaru,

Thank you for submitting your revised manuscript entitled "ciBAR1 loss in mice causes laterality defects, pancreatic degeneration and altered glucose tolerance". We would be happy to publish your paper in Life Science Alliance pending final revisions necessary to meet our formatting guidelines.

- please be sure that the authorship listing and order is correct
- please add a running title, summary blurb, keywords, and a category for your manuscript to our system
- please add the author contributions and a conflict of interest statement to the main manuscript text
- please use the [10 author names, et al.] format in your references (i.e. limit the author names to the first 10)

Figure Check:

- please add scale bars to Figure 1

A. FINAL FILES:

B. MANUSCRIPT ORGANIZATION AND FORMATTING:

Sincerely,

November 25, 2024

RE: Life Science Alliance Manuscript #LSA-2024-02916RR

Dr. Ken-Ichi Takemaru
Stony Brook University
Dept. of Pharmacological Sciences BST7-182, Nicolls Rd.
Stony Brook, NY 11794-8651

Dear Dr. Takemaru,

Thank you for submitting your Research Article entitled "ciBAR1 loss in mice causes laterality defects, pancreatic degeneration and altered glucose tolerance". It is a pleasure to let you know that your manuscript is now accepted for publication in Life Science Alliance. Congratulations on this interesting work.

DISTRIBUTION OF MATERIALS:

Again, congratulations on a very nice paper. I hope you found the review process to be constructive and are pleased with how the manuscript was handled editorially. We look forward to future exciting submissions from your lab.

Sincerely,
